# NarrativeXL: a Large-scale Dataset for Long-Term Memory Models

**Arseny Moskvichev**
Santa Fe Institute / Santa Fe, NM
arseny.moskvichev@gmail.com

**Ky-Vinh Mai**
University of California, Irvine / Irvine, CA
kyvinhm@uci.edu

## Abstract

We propose a new large-scale (nearly a million questions) ultra-long-context (more than 50,000 words average document length) reading comprehension dataset. Using GPT 3.5, we summarized each scene in 1,500 hand-curated fiction books from Project Gutenberg, which resulted in approximately 150 scene-level summaries per book. After that, we created a number of reading comprehension questions based on these summaries, including three types of multiple-choice scene recognition questions, as well as free-form narrative reconstruction questions. With 990,595 total questions, our dataset is an order of magnitude larger than the closest alternatives. Crucially, most questions have a known "retention demand", indicating how long-term of a memory is needed to answer them, which should aid long-term memory performance evaluation. We validate our data in four small-scale experiments: one with human labelers, and three with existing language models. We show that our questions 1) adequately represent the source material 2) can be used to diagnose a model's memory capacity 3) are not trivial for modern language models even when the memory demand does not exceed those models' context lengths. Lastly, we provide our code which can be used to further expand the dataset with minimal human labor.

## 1 Introduction

Although on paper many modern Large Language Models (LLMs) have maximal context lengths measured in tens of thousands of tokens, in practice, they often fail to access information plainly presented within those contexts [Liu et al., 2023] and their performance generally deteriorates as inputs grow larger.[1]

We believe that this issue is a consequence of the lack of supervised datasets that could be used to directly train extremely-long-context LLMs (as opposed to extrapolating from shorter sequences [Anil et al., 2022] which is prone to generalization errors). In our work, we create such a dataset. We capitalize on recent results showing that for tasks that do not require long-term memory, LLMs rival human labelers [Gilardi et al., 2023]; we use this "local" competence to create a massive datset (nearly a million questions in total) of extremely long-term memory problems (average context lengths from 54,334 to 87,051 words for different question types).

## 2 Language Modeling is Not Enough

In theory, optimal next-word prediction requires perfect memory of unlimited capacity, hence one might argue that supervised long-term memory datasets are unnecessary, given the abundance of unsupervised data. In practice, there are two considerations as to why Language Modeling alone might not be the best approach to train and test language models with extremely long context windows.

**First,** Language Modeling performance will likely see diminishing returns when the context window is increased. Many documents in popular unsupervised datasets are simply not long enough to benefit from contexts larger than ten thousand words. Additionally, for longer documents (e.g. fiction books), it is likely that remembering the last read chapter or two is nearly equivalent, in terms of the next word prediction quality, to remembering all the chapters read so far. It is possible that in some narratives, a given character or item might reappear after a long absence, but such cases are likely to happen only a few times per book, making the task extremely sparse and inefficient in training long-term memory models.

**Second,** language modeling does not offer a direct way to interpretably measure long-term memory capacity and performance. For example, we do often see improvement in perplexity when the effective context window is increased (e.g. [Dai et al.,

---

[1]This pattern, only recently observed in the literature, replicates on our data (s.f. subsection A.8).

2019]), but it is still difficult to measure and understand where exactly the improvement comes from and what kind of information is retained. One scenario could be that a longer context window helps a given model better understand lengthy philosophical treatises present in the dataset, which, in turn, allows the model to extrapolate such arguments in consistent and sound ways, resulting in lower perplexity. Alternatively, the model might simply be better at populating bibliography sections of such treatises, being able to copy the cited names from the main text into the bibliography using its long context.

We speculate, therefore, that in order for extremely long-context models to thrive, it is necessary to develop specialized supervised datasets that would address the limitations above. Creating such a dataset is the main contribution of our paper.

## 3  Existing datasets

Traditionally, long-term memory transformers were tested either on 1) artificial tasks (e.g. [Tay et al., 2020, Moskvichev and Liu, 2021, Weston et al., 2016]) or 2) language modeling (e.g. [Dai et al., 2019, Rae et al., 2019, Bulatov et al., 2022]).

Training or evaluation on supervised naturalistic long-term datasets is relatively rare. Until recently, creating such datasets in a brute-force manner had been prohibitively expensive, as it required tens of thousands of hours of human labor. There have been, however, creative workarounds taking advantage of existing resources. Notable examples include [Cohan et al., 2018] which used scientific papers from ArXiv and PubMed and their corresponding abstracts to create medium-term summarization data, and BookSUM [Kryściński et al., 2021] which scraped WebArchive to find summaries for project Gutenberg books. The SCROLLS dataset [Shaham et al., 2022] aggregated and curated a number of such datasets in order to create a long-text understanding benchmark. A recent Zero-SCROLLS dataset [Shaham et al., 2023] (concurrent with our work) expanded SCROLLS with two more sub-datasets tailored towards zero-shot evaluation.

In this context, it is especially important to discuss the NarrativeQA dataset [Kočiský et al., 2018] since this work is especially close to ours in its goals, scope, and structure[2].

In NarrativeQA, the authors employed crowd-source workers to create book and movie script understanding questions based on corresponding web-scraped summaries. While we highly resonate with the importance and motivation of their work, the dataset has a few crucial disadvantages.

1) Since all questions in NarrativeQA are written based on summaries alone, by construction, the dataset can not evaluate any reading comprehension that goes beyond knowing the summary. But, arguably, by reading a book, one gains detailed memories and understanding far exceeding what one could get by simply reading its summary. It seems highly desirable for any long-term reading comprehension dataset to reflect that.

2) The size of the dataset is limited by the number of pre-existing summaries available online, which restricts the dataset to approximately 1500 documents. Of those, only ∼400 are books, the rest being movie scripts, which are, generally, much shorter (hence the avarage document length in NarrativeQA is around 50,000, compared to 87,000 in our work). Overall, the NarrativeQA dataset contains ∼45,000 questions, which is good for evaluation, but might not be enough for training long-term memory models. Our dataset offers an order of magnitude more questions.

3) NarrativeQA does not offer a natural learning progression. All questions are asked at the end of the book/movie script which offers no natural curriculum for the model (e.g. learning to handle short retention questions first). In contrast, in NarrativeXL, our 726,803 multiple-choice questions cover a wide variety of memory demands – from a few thousand tokens to more than a hundred thousand. Importantly, our dataset also decouples memory demand (how long ago the relevant information was presented) from total context length (how much context was given in general), which makes it better suited for diagnosing and evaluating memory performance.

We discuss additional, subtler differences between NarrativeXL (our work) and NarrativeQA in subsection A.6. Overall, without diminishing the importance of NarrativeQA, we believe that the limitations above warrant the development of a new

---

[2]NarrativeQA is included as one of the subtasks in both SCROLLS and ZeroSCROLLS. In those datasets, documents from NarrativeQA offer by far the longest memory retention demands: the average number of words in NarrativeQA-subtask documents is 49,384, while the second-longest subtask offers only 10,839 words per document. Average document length in our data is 87,541, higher than in NarrativeQA since we use full-length books only.

| Dataset | Avg. Words | Documents | Task Items |
|---|---|---|---|
| BookSum [Kryściński et al., 2021] | **110,000** | 405 books | 405 book summaries |
| NarrativeQA [Kočiskỳ et al., 2018] | 50,000 | 1,572 (783 books, 789 screenplays) | 46,765 QA pairs |
| QMSum [Zhong et al., 2021] | 10,839 | 232 meetings | 1,808 query-summary pairs |
| QuALITY [Pang et al., 2022] | $\leq 6{,}000$ | 762 articles | 6,737 QA pairs |
| ASJ [Dangovski et al., 2021] | 5,975 | **50,134** academic papers | **50,134** press releases |

Table 1: Existing long-range dataset statistics. "Avg. Words" shows the average document length per task item.

| Narrative XL question type | Avg. Words | Documents | Task Items |
|---|---|---|---|
| Read along | 54,334* | 1500 Books | 726,803 (multiple choice) |
| Scene summary reconstruction | 87,051 | 1500 Books | 244,111 (freeform) |
| Hierarchical summary reconstruction | 87,051 | 1500 Books | 19,681 (freeform) |

Table 2: NarrativeXL (our contribution) dataset statistics. *Read along questions are asked before the book is fully read, reducing effective document length.

long-term reading comprehension dataset.

To the best of our knowledge, among naturalistic supervised Natural Language datasets, NarrativeQA [Kočiskỳ et al., 2018] is the only one coming close to our work terms of document lengths and the number of training items (s.f. subsection 7.2).

General statistics for a number of related datasets are provided in Table 1. When compared to NarrativeXL (see Table 2), it can be clearly seen that we offer a combination of context length and dataset size not covered by existing alternatives.

## 4 Methodology

Our general approach was to test long-term reading comprehension through book scene reconstruction and recognition. Since we want to encourage flexible memory representations rather than verbatim text memorization, instead of complete raw scenes, we used scene summaries. Our overall pipeline is illustrated in Figure 1 and is described below.

### 4.1 Data preparation

Raw books were downloaded from Project Gutenberg, with the boilerplate license information removed using a script. After that, we manually inspected each book, to remove 1) books that do not have an overarching narrative, such as short story collections, diaries, memoirs, published letters of prominent figures, and so on 2) author names, titles, table of contents, dedication, preface, addendum, glossary, translator's notes, and similar non-narrative information, 3) duplicate books. When

a given work has more than one volume, Project Gutenberg often has duplicate entries, storing both individual volumes and the same volumes merged into a single file. Keeping both versions of such books would have led to various dataset contamination issues. Overall, the goal of this stage was to select books that contain a single connected narrative and to remove irrelevant/contaminating information.

### 4.2 Summary creation

To obtain scene summaries, books were split into ∼3,000-symbol chunks (with 300-symbol overlap), and then each chunk was summarized using the GPT-3.5 API (the code, including prompt details, is provided at `https://github.com/r-seny/NarrativeXL`).

## 5 Question types

### 5.1 Read-along questions (multiple-choice)

Most reading comprehension datasets assume that their questions will be asked after the entire document is processed. In contrast, real-life linguistic activities are more "on-line". For example, one's understanding of a long dialogue or book does not suddenly materialize after the end of the text, rather, one's understanding continuously develops as the reading/talking proceeds.

To capture this property, we have constructed a large number of "read along" questions that are to be asked not at the end of the book, but rather at specific times as reading progresses. These questions are multiple-choice, in the form of "In what

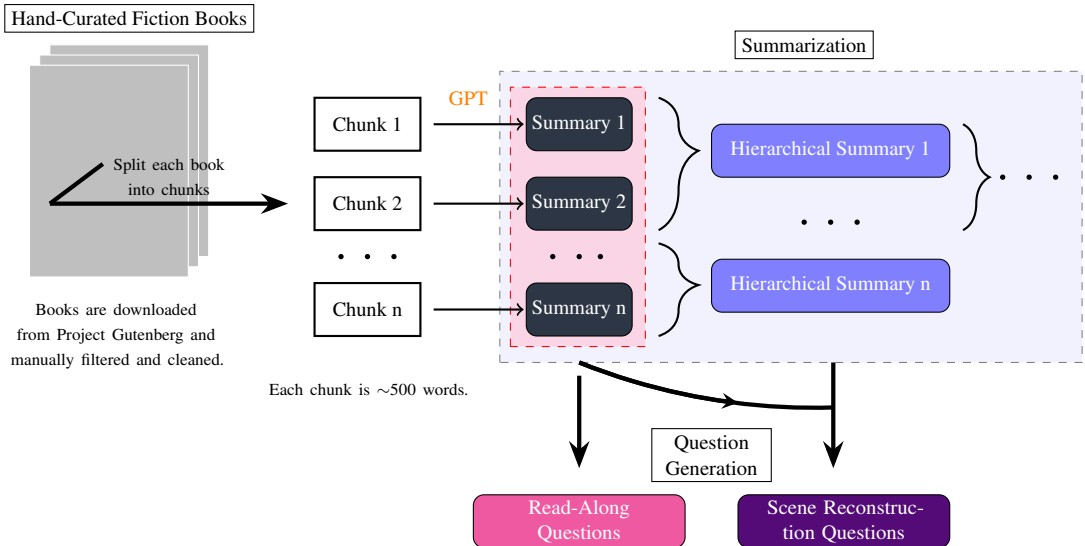

Figure 1: Data creation pipeline. Note that Read-Along Questions are generated only using summaries, while Scene Reconstruction Questions are generated using both summary and hierarchical summaries.

you've read so far, was there a scene where ...", after which a number of scene summaries are given, along with a "None of the above" option.

The true answer options are either true scene summaries from the book being read (see subsection 4.2), or "None of the above". Negative answer options are of three types: 1) Lookahead: scene summaries from the same book but from parts that have not been read yet at the time when the question is asked 2) Other book: scene summaries from other books (with character names substituted to match the true book) 3) Scene distortion: scene summaries that describe a similar setting but different events (generated using GPT-3.5). See Table 5.1 for illustrations.

Notably, the same question might have different answers depending on when it is asked, which, we hope, will discourage "overfit" solutions where a model simply memorizes all scenes in a given book. Additionally, each question has a clearly defined "memory load": how long ago was the target scene read. This endows the dataset with 1) natural curriculum learning opportunities 2) a simple way to measure any model's memory capacity by looking at its forgetting curve.

## 5.2 End-of-book summary reconstruction questions (freeform)

While our multiple-choice questions provide a controlled and interpretable way to measure memory performance, free-form answers might sometimes

provide a richer learning signal to the model. We, therefore, added "summary reconstruction" questions to our dataset, which take the following form: "Question: This partial book summary contains a number of errors. Rewrite it to accurately reflect the book you have read. [DISTORTED SUMMARY]", "Answer: [TRUE SUMMARY]", essentially mapping the rightmost column in Table 5.1 to the middle one. Here, true and distorted summaries are obtained using GPT-3.5 in the same way as in subsection 4.2 and subsection 5.1.

Additionally, we wanted to encourage models trained on our dataset to flexibly reason about the narrative on different time scales (not only on the scene level). To achieve that, we applied hierarchical summarization, obtaining true and false summaries that span different scales of text, starting with the scene level and up to the whole book summary.

Our distorted summaries are constructed to be plausible and generally fitting the book setting, while not being true to the book's events. We believe that this disentanglement of factual and stylistic knowledge will make our task better suited for training or fine-tuning long-term memory models than traditional next-word or masked-word predictions.

We also believe that the task is well-suited for testing Reading Comprehension as it requires 1) flexible knowledge of the overall narrative to recall the scene (or book part, for hierarchical summaries) structurally closest to the distorted one 2) detailed

| Summary distortion | |
|---|---|
| Book Snippet | Salt-air and dazzling society kept all idea of penance from this vivacious young person. It was queer that Sit Twickenham should be at the seaside, instead of at Brookfield, wooing; but a man's physical condition should be an excuse for any intermission of attentions. "Now that I know him better," wrote Adela, "I think him the pink of chivalry; and of this I am sure I can convince you, Bella, C. will be blessed indeed; for a delicate nature in a man of the world is a treasure. He has a beautiful little vessel of his own sailing beside us [...]" |
| True summary | The excerpt describes Adela and Arabella's different experiences during the yachting excursions. Adela is having a great time and writes to Arabella about the fun they are having. Arabella, on the other hand, is miserable and writes to Adela about the mundane daily events at Brookfield. The Hon. Mrs. Bayruffle accompanies the ladies on the yacht and Adela admires her social skills but realizes that society is not everything. The excerpt also touches on the idea that when people experience a great fall, they rarely indulge in melancholy until they can take it as a luxury. |
| False summary | The excerpt describes two friends, Adela and Arabella, taking a walk in the country-side. Adela is awestruck by the natural beauty around them and tells Arabella about the great time they are having. Arabella, however, is unimpressed and complains to Adela about the lack of proper civilization out here. The Hon. Mrs. Bayruffle joins the ladies for the walk and Adela learns a valuable lesson about the importance of solitude. The excerpt also touches on the idea that adversity reveals the true strength of a person's character. |

Table 3: True and distorted summary example. Crucially, we aimed to keep the setting the same, only changing the described events. This way, we hope to encourage models trained on our data to precisely remember book events rather than style, characters, or setting.

knowledge of the book events to properly reconstruct the original summary.

## 5.3 Expanding to other question types

To aid future research, along with the questions we have already generated, we also release the data generation scripts, true and false summaries for all scenes, and Named Entity substitution dictionaries (see subsection A.2). It is, therefore, easy to construct other tasks based on our data. It is also straightforward to expand our dataset if the application requires more data than what we provide.

## 6 Data Validation and Baselines

The primary goal of our dataset is to aid the development of long-term memory models. Therefore, our goal in data validation is not ensuring that our dataset can not be solved using alternative methods (e.g. retrieval-based), but rather making sure that our questions 1) can not be directly solved by language models without long-term memory 2) are diagnostic of the model's memory capacity 3) accurately represent the material on which they are based (i.e. our questions should be solvable).

### 6.1 Validating Read-Along Questions

#### 6.1.1 Testing for shortcut solutions

The first concern arises from the potential presence of shortcut solutions similar to those that have been recently plaguing the field (e.g. [Yang et al., 2020]). "Scene distortion" questions are especially susceptible: when such questions are generated, the "false" options might have subtle systematic differences from the true summary, as the true summaries are directly based on the source material, while the false summaries involve some "creativity" from GPT 3.5, which may have a particular style or inclination to fantasize about specific topics. On the other hand, "Lookahead" and "Other book" question types are symmetric by design (meaning that all answer options are generated in exactly the same way), and hence are not susceptible to such short-cuts.

To evaluate the extent of such biases (if any), we have fine-tuned BERT [Devlin et al., 2018] on "scene distortion" questions with no context (i.e. on answer options alone). We have used a subset of our data for which these options fit into BERT's context window of 512 tokens. The best of 5 runs

| | **Which of the following scenes was in the book?** |
|---|---|
| | **Answer Options** |
| 1) | In this book snippet, Theron, a minister, confronts Mr. Gorringe about receiving plants anonymously, unaware that they were smuggled in by him. Theron realizes that the plants were not intended for him, and Mr. Gorringe insinuates that Theron might have stolen the plants intended for someone else. Theron is hurt when Mr. Gorringe insults him and questions his integrity. The insult leaves Theron bewildered and speechless, unable to respond angrily. |
| 2) | In the book excerpt, a man suggests to a woman that she will go to Europe like American heiresses and marry a duke or nobleman and that princes would fight for her. She scoffs at the idea and proclaims her views on independence, stating that she belongs to herself and is not property for anyone to obtain rights over. She believes that women don't have to belong to somebody and that following the generally accepted views is not the only way to live as real human beings. Her companion is astounded by her words, finding them magnificent and likening them to the sensation he felt as a youth when he listened to the "Declaration of Independence." |
| 3) | Theron Ware accompanies Celia Madden to her home where he is surprised to find that the parlor is not on the ground floor and has to go up a broad, magnificent structure of stairs to get there. The room into which Celia leads him is not like any parlor Theron has ever seen. It is actually her workshop where she paints, models in clay, binds books, writes, draws, does carpentry, and all the things that make a mess which has to be cleaned up. Theron looks about him with undisguised awe. |
| | [ ...] |
| 6) | None of the above. |

Table 4: Question example. The model would have to determine the validity of each answer and select which summary accurately reflects something that happened in the text.

achieved an accuracy of 0.524 (with 6 categories, the random guess accuracy was at 0.167).

These results indicate that indeed, there are some idiosyncrasies that can help to distinguish between distorted and true summaries generated by GPT-3.5. Fortunately, they do not allow to unequivocally identify true summaries among the available distorted options, leaving ample room for long-term memory-based improvement. Additionally, this does not affect the effectiveness of scene reconstruction questions (subsection 5.2). Nevertheless, it is important to keep this imbalance in mind when interpreting long-term model memory performance on multiple-choice scene distortion questions.

### 6.1.2 Testing for memory impact

Apart from checking that it is not too easy for models to "cheat" on our data, it is also important to check that longer contexts do help models to answer our questions. Although at present, most LLMs can not fit complete books into their context, some of our questions (the ones asked early in the reading process, and having low "retention load") fall safely within their context lengths. We have

evaluated Claude v1.3 100k [3] and GPT-4 [OpenAI, 2023] models on a small subset of our data in a zero-shot manner. Each model received 60 questions with retention loads of no more than 8 scenes (∼4000 words), achieving overall accuracies of 0.53 and 0.783 for Anthropic and GPT-4 respectively[4]. This small experiment validates our data generation procedure, showing that knowing the relevant book context does help to answer the questions that we have designed. It also highlights the intuition that having a large enough context window is not equivalent to having perfect memory within the length of this context window.

### 6.1.3 Testing for adequacy

Apart from being balanced, we need our questions to accurately reflect book content. In order to test that, we have conducted a small-scale human study. Using a subset of our data, human participants[5]

---

[3] Anthropic (https://www.anthropic.com/)

[4] Which corresponds to 95% binomial confidence intervals of (0.4, 0.66) for Anthropic and (0.66, 0.88) for GPT-4.

[5] Participants were recruited from Amazon Mechanical Turk and compensated at $9.99 for a 40-minute study. We required US-based workers with a "master worker" qualification, 99% previous HIT approval rate, and at least 1000

were presented with randomly selected book scenes and two accompanying summaries, one true and one false, both generated by GPT-3.5. The task was to identify the true summary between the two. In total, 25 workers were recruited, being assigned 10 scenes each. Out of 250 total scenes, the workers correctly classified 238, which corresponds to 0.95 accuracy (95% binomial confidence interval [0.92, 0.97]). We would like to stress that this study served as a sanity check aiming to validate our data generation process, not to establish precise human performance benchmarks.

## 6.2 Validating end of book questions

To make sure that it is not possible to reconstruct original summaries from their corrupted versions without knowing the book, we fine-tuned the GPT3 Curie model on a sample of 700 summary reconstruction examples, with no book context. We also fine-tuned Longformer Encoder-Decoder (LED) [Beltagy et al., 2020] on our full training set. We measured the quality of the reconstructed summaries by comparing them to true (uncorrupted) summaries using ROUGE-$1_{F1}$, ROUGE-$2_{F1}$, ROUGE-$L_{F1}$[6] and BertSCORE-based F1[Zhang et al., 2019] metrics on a test set of 300 summary reconstruction pairs (full test set for LED). As a baseline, we used the similarity between corrupted and true summaries. This baseline is not trivial to beat since the corrupted summaries were constructed to have similar settings to the true ones. The results are presented in Table 5.

The fine-tuned models did not show a significant improvement in similarity scores over the corrupted summaries (fine-tuned GPT3, in fact, performed worse than the baseline). Manual inspection revealed that the models produced coherent reconstructions, changing some events in the corrupted summaries, e.g. "it was a gloomy rainy day" to "it was a sunny day" (see subsection A.3 for full examples). At the same time, as numerical evaluation shows (Table 5), the models failed to guess which events should be changed to reconstruct the true summaries.

Failure of the fine-tuned models suggests that it is not trivial to guess true summaries based on their corrupted versions. It might be, however, that our similarity metrics are simply not sensitive enough,

i.e. that fixing corrupted summaries to accurately reflect book events does not result in higher similarity with true summaries. To exclude this possibility, we ran an additional experiment using GPT-4 in a zero-shot manner to reconstruct scene summaries, with and without context (relevant part of the book on which the summary was based[7]). Knowing the book context resulted in significantly higher performance scores (paired t-tests show that the improvement of GPT-4 with context over the baseline is significant under all metrics).

Overall, these experiments show that guessing the true summaries based on their corrupted versions is not trivial. At the same time, knowing relevant raw book content improved performance, demonstrating that the task we designed is diagnostic of long-term memory capacity and the model's ability to use information within its context window.

## 7 Related work

### 7.1 Long-term memory transformers

There have been a number of notable efforts in developing new architectures and training procedures to introduce long-term memory into transformers. Brute-force approaches such as directly increasing the context window ([OpenAI, 2023]), along with works focusing on sparse attention mechanisms (see [Tay et al., 2022] for a review), often give good performance, but do not answer the question of how to transition from "very long working memory" to "long-term memory", as it is still not clear whether these context windows can be practically extended to capture human lifetime-scale experiences.

As recently shown in [Liu et al., 2023], many modern LLMs do not have full mastery over their claimed context lengths, and are often unable to use the information provided to them, especially as inputs grow longer. We observed similar results on our data (see subsection A.8). We believe that developing naturalistic supervised datasets that, like our contribution, focus on ultra-long contexts, will be crucial to overcome this issue.

Among methods that pursue alternative memory mechanisms rather than larger context windows, one line of research explores knowledge-base-like storage of previous interactions [Lewis

---

previously completed HITs.

[6]ROUGE scores were calculated using this Python library: https://pypi.org/project/rouge-score/

[7]Experiments in this section did not use hierarchical summaries, since for them, it would be impossible to fit relevant book content into GPT3/4 context windows.

|  | $R1_{F1}$ | $R2_{F1}$ | $RL_{F1}$ | $BertSCORE_{F1}$ |
|---|---|---|---|---|
| Baseline | .522 | .261 | .408 | .906 |
| LED (fine-tuned) | .53 | .26 | .40 | .90 |
| GPT-3 (fine-tuned) | .504 | .236 | .384 | .9 |
| GPT-4 no context | .512 | .250 | .396 | .904 |
| **GPT-4 with context** | **.576** | **.295** | **.422** | **.913** |

Table 5: Summary reconstruction performance. Baseline is obtained by comparing corrupted and uncorrupted summaries. All models were trained on a subset of our data, except for LED which was trained on the complete dataset. Overall, GPT-4 with original book context performs best, while GPT-3 and LED finetuned to correct summaries without knowing the book fail to find any shortcut solutions and performed near baseline. GPT-4 with no context also fails to "guess" original book events. Overall, these results indicate that modern LLMs can not answer summary reconstruction questions well without knowing the book context. See subsection A.7 for a version of this table with confidence intervals.

et al., 2020]. Another approach is to endow transformers with a distributed memory representation that it learns to update. Thus, [Moskvichev and Liu, 2021] proposed a practical end-to-end way to train transformer-like architectures to update a distributed memory state, while [Rae et al., 2019] proposed a way to compress past memories while separating memory state gradients from the main training objective. Lastly, model editing can also be seen as a form of long-term memory: this fruitful line of research focuses on incorporating new information directly into the model's weights using gradient updates [Zhu et al., 2020].

Overall, without making a prediction on which long-term memory mechanisms will be most successful in the future, we hope that our dataset will help in training and evaluating such long-term memory models.

### 7.2 Long-term memory datasets

There are a number of datasets that can be used to evaluate long-term memory models, but most of them are substantially different from our approach. In section 3 we focused on the differences between our dataset and NarrativeQA [Kočiskỳ et al., 2018] (which is closest to our work in scope and goals). Here we offer additional comments on how our work compares to a few other benchmarks.

A work concurrent with ours, the Zero-SCROLLS dataset [Shaham et al., 2023] combines a number of previous datasets together with two newly proposed tasks to create an evaluation suite for zero-shot long-term memory tasks. Compared to our work, ZeroSCROLLS dataset is of much smaller scale, with 10 datasets (two new, 8 adopted from previous work) of 500 or fewer examples each. Its diverse set of tasks makes ZeroSCROLLS

well-suited for long-term memory evaluation, but the scale of the dataset is not sufficient for training. Moreover, the longest documents in the ZeroSCROLLS dataset still come from NarrativeQA (average document length (50,000) words, with second-longest subtask (QMSum [Zhong et al., 2021]) at a modest 10,839).

Other relevant works also fall into substantially different data regimes compared to our contribution. For example, QuALITY [Pang et al., 2022] offers 6,737 carefully hand-crafted questions based on 762 documents of up to 6000 words. In contrast, our dataset was less carefully curated but is of a much larger scale, with 726,803 multiple choice questions, and 263,792 scene reconstruction questions based on 1,500 documents averaging 87,051 words long. The Automatic Science Journalism dataset [Dangovski et al., 2021] offers a larger number of documents (50,134), but their average length is 5,975 words, which might not be enough to push modern LLMs out of their "comfort zone" in terms of memory demands.

The BookSum [Kryściński et al., 2021] dataset offers a collection of web-scraped book-, chapter-, and paragraph- level summaries. Compared to our dataset, BookSum is smaller (based 405 full-text documents, only 120 of which are full-length novels) and is focused exclusively on summarization. Given its small number of texts and summaries, the BookSum dataset is not well suited for training long-term memory models, but can be useful for benchmarking.

Overall, our dataset fills an important role by offering large quantities of extremely long-range reading comprehension tasks that, we speculate, will be sufficient to train (and not only evaluate) long-term memory reading comprehension mod-

els. What is also important is that our dataset offers a natural learning progression by having a range of questions with varying memory demands. This adds a natural curriculum learning opportunity, where a given model can first learn to answer shorter-context questions, and gradually transition to more challenging ones. Overall, our work fills an important and under-explored niche, complementing a number of recently proposed smaller-scale datasets.

## 8 Limitations

It is likely that many of our questions can be answered using relatively simple Information Retrieval approaches (IR), e.g. by reversing our data generation process and scanning each scene in a book with a GPT-like model. We would like to stress that this does not undermine the purpose of our study, similarly to how the existence of simple hard-coded solutions to some of the tasks in the Long Range Arena challenge [Tay et al., 2020] did not negate the impact of that work. We aimed to create a naturalistic dataset that could be used to train and evaluate language models with long-term memory capacity. It is possible that any such dataset can be solved with alternative Information Retrieval methods, since IR can be interpreted as having perfect memory (unrestricted access to the whole document from which the information should be retrieved). Nevertheless, there is a need for non-IR-based long-term memory models, and we believe that our dataset offers exactly what is needed to train and evaluate such models.

Data contamination. It is impossible to control which books are included in any given LM's training set, and being exposed to a given book in advance might aid performance [Chang et al., 2023]. We do not claim to fully resolve the issue, but do take steps to ameliorate it by removing book titles and author names, changing the named entities, and basing questions on scene summaries, rather than on raw scenes. With these measures, we hope to make it harder for models to map books they are reading to something they might already know. Additionally, our read-along questions give different answers depending on when they are asked. This makes it necessary for any model to rely on its memory to track the reading progress even if it was already exposed to the book before. In future work, it might be beneficial to paraphrase the books in our dataset to further mitigate data contamination.

Lastly, we do not directly show that our dataset will improve ultra-long-context LLMs or LLMs with long-term memory mechanisms. Instead, we focused our energy and resources on validating our data and making sure that our tasks test what they were designed to test: information retention and access in texts of extreme length. In the future, we hope to see new LLMs developed and evaluated using our data.

## 9 Conclusion

We have proposed a new reading comprehension dataset for extreme-long-context LLM training and evaluation. Based on full-length books, it offers higher average document lengths and an order of magnitude more examples than any of the available alternatives (e.g. [Kočiský et al., 2018, Kryściński et al., 2021]). We have conducted four data validation experiments, demonstrating that our data accurately reflects the source material and is diagnostic of long-term memory performance. Additionally, our method allows us to further expand the dataset at a very low cost, making it feasible, for example, to label all books in the Gutenberg corpus at a price realistic for many academic and industry organizations.

We hope that in the future, the data we provide will aid in training and evaluation of ultra-long context LLMs. We also hope that our work inspires further research into creating high-quality datasets for language understanding on even more extreme time scales (millions of tokens and more), as ultimately, it is that timescale that is most representative of human linguistic experience.

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

# A Appendix

## A.1 Code and data availability

The code and data are be available at `https://github.com/r-seny/NarrativeXL`.

## A.2 Named entity substitution

Due to data contamination, a model trained on our data might "know" some of the books since its pre-training stage. To thwart such models' ability to rely on this knowledge, we identify and randomize character names in each book (similarly to how it was done in [Kočiskỳ et al., 2018]). It is especially important for the "other book" decoy questions (see subsection 5.1), as we want to avoid shortcuts where scene summaries from other books can be identified simply by looking at the named entities mentioned in them. We used spacy[8] library to identify named entities. We took extra care to substitute different versions of the same name in a consistent manner (e.g. separately keeping track of each part in multi-word names). We release both substituted and non-substituted dataset versions so that in the future, a different named entity detection engine can be used to improve name substitution quality.

## A.3 Reconstruction examples

A few illustrative examples for GPT-3 (curie) fine-tuned model reconstructions. These prompts were manually created to explore model behavior. In general, the model picked up on the idea that some details should be changed, but, as our evaluation shows, the model, on average, failed to guess which details should be changed to reconstruct the original (uncorrupted) summaries.

**Prompt 1:** It was a gloomy rainy day. Patricia went for a walk with her dog, and met Jack, who was jogging in the park. They exchanged casual conversation, then went on their separate way. Jack's lack of attention left Patricia wondering whether Jack actually had any feelings for her.

**Reconstruction 1:** It was a sunny day. Patricia went for a walk with her dog, and met Jack, who was sailing his boat on the lake. They exchanged casual conversation, then went on their separate way. Jackś lack of attention left Patricia wondering whether Jack actually had any feelings for her.

---

[8] https://spacy.io/

**Prompt 2:** After three days in sweltering Hawaiian midsummer weather, Mary and John gave up on any attempts at outdoors entertainment and stayed in their hotel, mostly watching TV and playing tabletop games. Although it felt like a wasted opportunity, this experience gave them confidence that they can be happy in any circumstances.

**Reconstruction 2:** Mary and John are on a vacation in Hawaiian midsummer weather, and they are struggling to stay cool. They try to find some relief from the heat by going out for a walk, but the humidity makes it difficult to breathe. They end up spending most of their time indoors, trying to find ways to stay cool. They try to find some relief from the heat by going out for a swim, but the beach is too rocky for swimming. They end up spending most of their time in the hotel's pool, trying to stay cool. They also try to find ways to stay cool, such as taking cold baths, drinking cold lemonade, and wearing as many clothes as possible. Despite their efforts, they still feel hot and uncomfortable.

**Prompt 3:** Walther, the famous detective, inspected the crime scene. His fifteen years of experience gave him confidence that he won't miss a single important detail. This time, however, this confidence proved faulty, and it was his young partner, Taylor, a criminologist in training, who picked up on the key clue: a piece of cheese with a bitemark that indicated the the suspect had a crooked tooth.

**Reconstruction 3:** Walther, the famous detective, inspected the crime scene. His fifteen years of experience gave him confidence that he won't miss a single important detail. This time, however, this confidence proved faulty, and it was his young partner, Taylor, a criminologist in training, who picked up on the key clue: a piece of cheese with a bitemark that indicated that the suspect had a straight tooth.

## A.4 Additional dataset statistics

Here we provide additional dataset statistics. For read-along questions, the average question length (without context) is 499.51 words, with a standard deviation of 54.78 and an interquartile range of 462 to 534. The question length is relatively high since the questions ask to choose among a number of candidate scene summaries.

The average context length for these questions is 54,335 (sd=48,204; IQR[21,932-71,294]) words. Notably, our data provides a highly favorable fea-

ture: an option to calculate a "memory retention demand", i.e. we know not only the overall context length preceding every read-along question, but also where in this context the relevant information is located. The average memory demand for these questions is 31,931 (sd=36,597 words, IQR[7,596-43,536]). The standard deviations are relatively high since our dataset provides an abundance of different context lengths and memory demands. This should allow for natural curriculum learning opportunities since our dataset gives a natural progression from short to long context questions. It will also allow to systematically evaluate model performance in different context length regimes.

For summary reconstruction questions, out of 262,292 summary reconstruction questions, 244,111 are non-hierarchical (i.e. directly reconstructing scene summaries). Then, there are 15,086 first-level hierarchical summaries, 2391 second-level, 703 third-level, and 1 fourth-level hierarchical summary. Generally, the hierarchy level (how many iterative summarizations a book went through) depended on book length. The average context for summary reconstruction questions length was=87,051 (sd=42,682, IQR[59,459-98,565]) with the average question (distorted summary) length of 90.54 (std=32.5, IQR[76-106]) and the average answer (true summary) length of 104.98 (sd=41.05, IQR[83-116]).

It is worth noting that there is a slight discrepancy in average document length between our training and test sets (90,321 vs 80,512 average lengths in train and test respectively). It is a result of random book/author sampling. Some of the more wordy authors happened to be placed in the training set (we generally limited the number of books from one author, but also avoided having books from the same author in different data splits).

Additionally, it is important to mention that the books that we selected for our dataset partially overlap with PG-19 [Rae et al., 2019]; 53.6% of our books are also found in PG19. Information about which books in our dataset are also in PG19 is released along with our code and data. This way, it will be possible for models trained on our data to see if there is a difference in memory performance for books included in PG19 vs the rest.

We believe that this might be better than filtering out PG19 entirely. First, PG19 books are a good data source, second, having PG19 books in our data would help to better diagnose the source of memory improvements. For example, if a new long-term memory model only performs well on books it already knows from PG19, it would indicate that the memory mechanism needs substantial improvement, while if they perform nearly equally on PG19 and non-PG19 books, it would indicate the actual ability to handle extreme-long-context documents. In other words, since it is nearly impossible to avoid data contamination, it might be better to have the option to systematically evaluate its impact.

## A.5 Costs

Using our pipeline, processing a single book costs $\sim$ \$0.15 to create scene summaries, $\sim$ \$0.15 to create false scene summaries. The total cost of $\sim$ \$0.3 per book is two orders of magnitude less than what can be achieved with crowdsourced human labor (assuming a very fast reading speed of 5 hours per book and a moderate pay of \$10/hour). In our case, initial book filtering (removing non-narrative books) was done manually, but with each book taking less then a minute to skim, this work can be outsourced at a very low cost.

## A.6 Additional discussion on NarrativeQA

Here we would like to expand our discussion in section 3 about the differences between our contribution and NarrativeQA, exploring two additional, more subtle points.

First NarrativeQA dataset selected only books and movie scripts that had corresponding Wikipedia plot summaries (according to the paper, it was the difficulty in finding these summaries that ultimately limited the dataset size [Kočiský et al., 2018]). In practice, such summaries are usually present only on Wikipedia pages that cover highly popular movies and books. Unfortunately, popular books, including their key events, plot twists, and development, are likely to be extensively discussed on various review websites, social networks, and so on. Thus, any LLM trained on unsupervised web-scraped data (as most LLMs are) is likely to have extensive knowledge about these books and movies. Our dataset does not solve this issue (since we still rely on publicly available books), but mitigates it, as we do not bias our dataset towards popular books. In fact, many of the books in our dataset have no Wikipedia pages, reviews, or summaries we could readily find online.

Lastly, we have manually filtered 1500 Project Gutenberg books that our dataset is based on. In

that process, it became evident that many books, especially highly impactful ones, often included prefaces discussing the contents of the book, and story blurbs and summaries. The books in the NarrativeQA dataset were not filtered/processed beyond making sure that web-scraped summaries matched the books. This further exacerbates the previous issue, indicating that the dataset might be, to an extent, "self-contaminated".

### A.7 Summary reconstruction extra information

Table Table 6 provides uncertainty estimates for Table 5.

### A.8 LLM performance for different context lengths

To make sure that our dataset is capable of diagnosing model memory competence across variable context lengths and memory loads, we performed two additional experiments, one with claude-instant-1.2 and one with claude 2.0 (both supporting up to 100k token contexts).

We sampled two groups of "read along" questions: "short context", designed to be asked after 8 book scenes (average context length 4,077 words), and "long contex", asked after 100 book scenes (average context length 50,977 words). For claude-instant, we sampled 298 questions (150 short, 148 long), for claude 2.0, due to its higher cost, 148 (75 short, 73 long). Results followed the same pattern for both models; for brevity, we report Claude 2.0 results only, which is the stronger of the two models.

Average accuracy for short context questions was 0.51, much higher than for long context questions 0.26 (with 6 answer options, random guess accuracy was 0.167). Statistically, there was a strong negative correlation between context length indicator variable (0 for short, 1 for long) and performance (Spearman's rank correlation: -0.25, p=0.002). To make sure our results do not differ based on the chosen statistical analysis method, we replicated the analysis using a contingency table chi-square test, obtaining a chi-square statistic of 9.48 and a p-value 0.002 (8.4719 and p = 0.004 if Yates correction is used).

Overall, we see strong evidence that modern LLM performance deteriorates on longer contexts, and that extreme-long-context questions remain a challenge even for models that "in theory" support such context lengths. This replicates recent results reported in [Liu et al., 2023] and highlights the importance of our work.

### A.9 Prompts

In this section, we provide prompts that we used with GPT-3.5 to create our book summaries, hierarchical book summaries, and their corrupted versions. We also provide Anthropic and GPT-4 prompts for our zero-shot data validation experiments.

#### A.9.1 GPT-3.5 summarization

For (non-hierarchical) summarization, we used the following prompt structure:

> System: 'You are a helpful assistant that summarizes book snippets. Begin your answer with a ### BEGIN ANSWER ### tag.'
>
> User: 'Summarize the following book excerpt: "<BOOK CHUNK>". Start your answer with a ### BEGIN ANSWER ### tag.'

We observed that having the "### BEGIN ANSWER ###" tag allowed to avoid "helpful" comments from the model, such as "Sure, let me summarize this book snippet for you." The summary was parsed from the response as everything after the "### BEGIN ANSWER ###" tag.

Sometimes, the model appended "### END ANSWER ###" to its output, if that happened, that tag was removed. If the model failed to add the begin answer tag, we changed the "User" part of the prompt to a more forceful version and re-queried the model:

> User: 'Summarize the following book excerpt: "<BOOK CHUNK>". Make sure to begin your answer with a "### BEGIN ANSWER ### tag!'

If the model failed ten times in a row, we marked the chunk as failed ("unsummarizable"), and it was excluded from subsequent question generation. Such cases were exceedingly rare.

#### A.9.2 GPT-3.5 creating false summaries

> System: "You are a helpful assistant that changes book snippet summaries. Begin your answer with a ### BEGIN ANSWER ### tag."
>
> User: 'Take the summary below and rephrase it in such a way that the described events are

| | $R1_{F1}$ | $R2_{F1}$ | $RL_{F1}$ | $BertSCORE_{F1}$ |
|---|---|---|---|---|
| Baseline | .522 (.507, .537) | .261 (.240, .282) | .408 (.388, .427) | .906 (.903, .909) |
| LED-base (fine-tuned) | .52 | .26 | .40 | .90 |
| LED-large (fine-tuned) | .53 | .26 | .40 | .90 |
| GPT-3 (fine-tuned) | .504 (.489, .518) | .236 (.217, .255) | .384 (.367, .402) | .9 (.897, .903) |
| GPT-4 no context | .512 (.497, .527) | .250 (.23, .271) | .396 (.378, .415) | .904 (.901, .907) |
| **GPT-4 with context** | **.576 (.564, .588)** | **.295 (.287, .313)** | **.422 (.406, .438)** | **.913 (.911, .915)** |

Table 6: Summary reconstruction performance. The baseline was obtained by comparing corrupted and true summaries. Additionally, 95% confidence intervals are given in parentheses (based on performance variation for 300 samples in the test set. LED models were evaluated on our full test set). Overall, GPT-4 with original book context performs best, while GPT-3 and LED models finetuned to correct summaries without knowing the book fails to find any shortcut solutions and perform near baseline. Unsurprisingly, GPT-4 with no context also fails to "guess" original book events.

no longer the same, even though the setting remains the same. Keep your summary to the same length. Start your answer with a "### BEGIN ANSWER ###" tag. \nInital summary:\n"<SUMMARY>"'

### A.9.3 GPT-3.5 creating hierarchical summaries

System: 'You are a helpful assistant that summarizes book scene summaries. Begin your answer with a ### BEGIN ANSWER ### tag.'

User: 'Describe the events in following scene summaries into one plot summary: "<SUMMARY 1, ... SUMMARY N>". Make sure to list the key events and plot developments. Make sure to begin your answer with a ### BEGIN ANSWER ### tag!'

Here, summaries 1 to N are obtained by concatenating lower-level summaries. The maximum concatenation length was set to 10000 symbols.

### A.9.4 GPT-3.5 creating false hierarchical summaries

System: 'You are a helpful assistant that changes book snippet summaries. Begin your answer with a ### BEGIN ANSWER ### tag.'

User: 'Take the summary below and rephrase it in such a way that the described events are no longer the same, even though the setting remains the same. Keep your summary to the same length and keep your summary structure the same. Start your answer with a "###

BEGIN ANSWER ###" tag. \nInital summary: \n"<INITIAL HIERARCHICAL SUMMARY>"'

### A.9.5 GPT-4 and Anthropic multiple-choice question answering test

For Anthropic's Claude model, we used the following prompt structure:

"<anthropic.HUMAN_PROMPT> You will read a large part of a book, after which I will ask you questions about what you've read. Book part:\n<BOOK CONTEXT> \nQuestions:<QUESTIONS>\nWrite only the numerical answers to the corresponding questions, separating them by commas. For example, '1,3,4'. Begin your answers with a ### BEGIN ANSWER ### tag. <anthropic.AI_PROMPT>"

For the "User" portion of the GPT-4 prompt, we used the same prompt without <anthropic.HUMAN_PROMPT> and <anthropic.AI_PROMPT> which are specific to Anthropic API. For the "System", we used "You are a helpful assistant that reads large book snippets and answers questions about those snippets. Begin your answer with a ### BEGIN ANSWER ### tag."

Here, "Questions" are formatted in the same way as in Table 4. Models received three questions at a time.

### A.9.6 GPT-4 summary reconstruction prompts

The "System" portion of the prompt was the same for with- and without- context experiments: "You are a helpful assistant that corrects innaccurate book part summaries. Begin your answer with a ### BEGIN ANSWER ### tag."

For the experiment without context, the "User" part of the prompt was

"You will read a summary that covers a part of a book, but misrepresents events in it. Correct it so that it accurately represents the book events:\nIncorrect summary: '<FALSE SUMMARY>'\n\nYou will not have access to the original book, but try to do your best.\n\nWrite only the corrected summary, do not explain your solution. Keep the same summary structure.\nBegin your corrected summary answer with a ### BEGIN ANSWER ### tag.

For the experiment with context, the "User" part of the prompt was

"You will read a summary that covers a part of a book, but misrepresents events in it. Correct it so that it accurately represents the book events:\n\nIncorrect summary: '<FALSE SUMMARY>'\n\nOriginal book snippet:\n'<RELEVANT BOOK SNIPPET>'\n\nWrite only the corrected summary, do not explain your solution. Keep the same summary structure.\nBegin your corrected summary answer with a ### BEGIN ANSWER ### tag."