# OpenReview forum: "NarrativeXL: a Large-scale Dataset for Long-Term Memory Models"
_EMNLP/2023/Conference — EMNLP 2023 Findings_

### Official Review · Reviewer_ms7Q · 2023-07-27

**Soundness:** 3

**Excitement:**

3: Ambivalent: It has merits (e.g., it reports state-of-the-art results, the idea is nice), but there are key weaknesses (e.g., it describes incremental work), and it can significantly benefit from another round of revision. However, I won't object to accepting it if my co-reviewers champion it.

**Paper Topic And Main Contributions:**

The paper collects a new dataset which can evaluate language model's long-term memory capability. The corpus is from Project Gutenberg and manually inspected to ensure quality. Then, the paper proposes two question types, read-along questions and summary correction questions to evaluate model's memory capacity and global understanding. The answers are generated by GPT 3.5 API and the dataset is easy to construct other questions.

**Questions For The Authors:**

Question A: Is your corpus from PG is filtered with existed PG19 datasets? I concern that almost all of the pre-trained models are using PG19. However, using as the training corpus doesn't affect the evaluation for long-term capacity. But if some models give more pre-training corpus percentage on existing PG datasets, if may contribute to better results.

Question B: Are the answers generated by GPT-3.5 good enough to be a dataset source? For example, some comparisons with hand-labeled methods may explain a lot. I don't doubt the quality of GPT series, but it can't be the best model 5 years later.

**Reasons To Accept:**

The dataset's building pipeline is novel and adequate for their claims. The question design is a smart way to evaluate real long-term performance.

**Reasons To Reject:**

I'm not sure that using GPT series to generate answers is a fair method for comparisons with other models. It may cause that GPT is always better than other models even though there are great improvements in open-source community.

**Reproducibility:**

5: Could easily reproduce the results.

**Reviewer Confidence:**

5: Positive that my evaluation is correct. I read the paper very carefully and I am very familiar with related work.

---

> ### Author Rebuttal · Authors · 2023-08-29
>
> Thank you for your review and for your comments! Below we would like to address some of your concerns.
>
> __Q: “I'm not sure that using GPT series to generate answers is a fair method for comparisons with other models. It may cause that GPT is always better than other models even though there are great improvements in open-source community.”__
>
> A: This is a good point, and we now mention it in the limitations. We believe, however, that it’s not as critical as it might seem, for a few reasons:
>
> 1. We used GPT3.5 only to generate local scene summaries and their distorted versions, and we then used human labelers to validate that the summaries accurately represent original text. Generally, since short-scene-summarization performance is essentially at human level for most modern LLMs, we hoped that it would lower the impact of  specific model choice for summary generation. In the preliminary stages of our study, we tried a number of open-source models for summary generation, but GPT3.5 offered the best quality to resource ratio.
> 2. Given how our data was generated, the only potential advantage for the GPT-family of models might come from stylistic differences in how GPT3.5 writes its local summaries. To see whether GPT3.5 has an advantage because of it, during the rebuttal period, we conducted an additional experiment for section 6.2 (Table 3), fine-tuning two openly available Longformer-Encoder-Decoder models (base and large versions), like we did with GPT3.5. They performed slightly better than GPT3.5 and GPT4 with no context, but worse than GPT4 with context. This again suggests that it’s the memory performance and not difference in writing style that determines model performance on our data.
> 3. We generated long-term memory questions based on GPT-generated local summaries, but GPT3.5 does not actually have the memory capacity to answer the resulting questions. Even if there is some advantage to GPT3.5 because it was involved in data generation, this specific model won’t be able to capitalize on that.
> 4. It is not clear how similar the GPT-based family of models are, compared to other LLMs. That is, it’s not clear whether GPT4 is similar enough to GPT3.5 to get any advantage on our dataset, compared to, say Claude or any of the open-source models.
>
> Most importantly, we see the main impact of our work in that it can be used to train and diagnose long-term memory models. Each question has a known memory retention demand, showing how much context the model needs to remember to answer it. We find that there is no good dataset for this at present. Even if the evaluation was slightly biased towards GPT models overall because of, say, stylistic differences in summary writing, it should not affect the shape of the forgetting curve.
>
> __Q: “Is your corpus from PG is filtered with existed PG19 datasets? I concern that almost all of the pre-trained models are using PG19. However, using as the training corpus doesn't affect the evaluation for long-term capacity. But if some models give more pre-training corpus percentage on existing PG datasets, if may contribute to better results.”__
>
> A: Thank you for pointing this out! We have not filtered PG19 dataset out, but we did now calculate the overlap percentage with PG19. We find that 53.6% of our books are also found in PG19.
>
> We will release information about which books in our dataset are also in PG19. This way, it will be possible for models trained on our data to see if there is a difference in memory performance for books included in PG19 vs the rest.
>
> We believe that this might be better than filtering out PG19 entirely. First, PG19 books are a good data source, second, having PG19 books in our data would help to better diagnose the source of memory improvements. For example, if a new long-term memory model only performs well on books it already knows from PG19, it would indicate that the memory mechanism needs substantial improvement, while if they perform nearly equally on PG19 and non-PG19 books, it would indicate actual ability to handle extreme-long-context documents.
>
> In other words, since it is nearly impossible to avoid data contamination, it might be better to have the option to systematically evaluate its impact.
>
> __Q: “Are the answers generated by GPT-3.5 good enough to be a dataset source? For example, some comparisons with hand-labeled methods may explain a lot. I don't doubt the quality of GPT series, but it can't be the best model 5 years later.”__
>
> A: This is a great question. We conducted extensive experiments to make sure that our method results in good questions. It is important to remember that GPT3.5 only generated local summaries and their distorted versions, not full questions and their answers. Therefore, we only need local summary performance to be good. As described in section 6.1.3, we conducted a human-participant study to evaluate the quality of these local summaries.
>
> During the rebuttal period, we expanded this study to 25 highly experienced MTurk participants and 250 total summary evaluations. Participants were given a book snippet and a pair of GPT3.5-generated summaries (one accurate and one misrepresenting the data). Participants were able to identify the correct summary in 238 out of 250 cases, which corresponds to 0.95 accuracy (95% binomial confidence interval [0.92, 0.97]). It indicates that our data generation process was adequate for the task.
>
> Overall, we only apply GPT3.5 to a local task at which it is already extremely good. This, we hope, would ensure the longevity of our data even as better overall models are developed. That being said, we would be happy to see our methodology extended in the future using other, more advanced models.
>
> Thank you again for your review and your comments. We hope that our response mitigates some of your concerns.

---

### Official Review · Reviewer_wT87 · 2023-07-29

**Soundness:** 4

**Excitement:**

4: Strong: This paper deepens the understanding of some phenomenon or lowers the barriers to an existing research direction.

**Missing References:**

Dangovski, R., Shen, M., Byrd, D., Jing, L., Tsvetkova, D., Nakov, P., & Soljačić, M. (2021, May). We Can Explain Your Research in Layman's Terms: Towards Automating Science Journalism at Scale. In Proceedings of the AAAI Conference on Artificial Intelligence (Vol. 35, No. 14, pp. 12728-12737). -- dataset with long scientific documents and their long press releases for abstraction summarization (for section 7.2)

**Paper Topic And Main Contributions:**

The authors address the problem of lack of training and evaluation data for long-context LLMs. The paper presents a novel long-document dataset that consists of well-curated books from Project Gutenberg, which are chunked and fed into GPT 3.5 to create questions and labels that accompany the story. Through the process of hierarchical summarization, the authors create questions and labels that correspond to the whole book. The authors study the merits of their work by training GPT3 and evaluating Claude and GPT4 on the task in a variety of experimental settings.

**Reasons To Accept:**

1. The type of dataset is novel, and it addresses some of the issues with previous long-document dataset (e.g. context corruption, training data corruption, etc.)

2. The evaluation experiments are meaningful and support the addition of that dataset for evaluation.

3. The methodology could be used by the community to be scaled to other datasets.

**Reasons To Reject:**

1. The actual statistics of the dataset are missing from the paper. We need to see a table with mean and stddev of the length of the prompts, as well as the targets.

2. While the authors claim that they are mitigating one of the problems with current LLMs which states that they are not trained on log-document data, the authors do not train an LLM on their dataset. While I understand that the resources to accomplish that would be very high, it remains a limitation given the current presentation of the paper.

**Reproducibility:**

4: Could mostly reproduce the results, but there may be some variation because of sample variance or minor variations in their interpretation of the protocol or method.

**Reviewer Confidence:**

3: Pretty sure, but there's a chance I missed something. Although I have a good feel for this area in general, I did not carefully check the paper's details, e.g., the math, experimental design, or novelty.

---

> ### Author Rebuttal · Authors · 2023-08-29
>
> Thank you very much for your time reviewing our paper and for your comments. Please note that, along with this individualized rebuttal, we have also added an official comment, summarizing our rebuttal contributions to all reviewers.
>
> Below we respond to each of your concerns and suggestions.
>
> __Q: "The actual statistics of the dataset are missing from the paper. We need to see a table with mean and stddev of the length of the prompts, as well as the targets."__
>
> A: We deeply apologize for not including overall dataset statistics in our original submission. We have now added large-scale dataset information comparing our dataset to existing alternatives in terms of the total number of documents, questions, and so on (please see our official comment). Additionally, the following information (and more detailed descriptive statistics and distribution visualizations) will be included in the appendix section of our paper:
>
> | Question type                        | Context length                  | Question length (words)        | Answer length (words)               |
> |--------------------------------------|------------------------------------------|-------------------------------------|-------------------------------------|
> | Read-along (multiple choice)         | mean=54334 (sd=48204, IQR[21932, 71294])  | mean=499.5 (sd=54.8, IQR[462, 534]) | 1 (multiple choice)                 |
> | End of book (summary reconstruction, free-form) | mean=87051 (sd=42682, IQR[59459, 98565]) | mean=104.4 (sd=41.7, IQR[83, 116])  | mean=90.0 (std=33.01, IQR[76, 106]) |
>
> *The context length is shorter for read-along questions since they are asked before the book is fully read.
>
> __Q: "While the authors claim that they are mitigating one of the problems with current LLMs which states that they are not trained on log-document data, the authors do not train an LLM on their dataset. While I understand that the resources to accomplish that would be very high, it remains a limitation given the current presentation of the paper."__
>
> A: We agree with this criticism. We now acknowledge it in the limitation section of our paper. We also re-worked the phrasing of our paper throughout, making it more clear that we speculate (rather then decisively prove) that our dataset should help mitigate long-term memory problems of modern LLMs.
>
> Q: Missing reference
>
> A: Thank you very much for pointing this work to us. We have included a short discussion of Dangovski, R. et al. (2021) to section 7.2, and we have also added summary statistics for this dataset to the table that now summarizes other work addressing similar issues as ours (please see our "rebuttal summary" official comment). As can be seen from the table, we believe that this dataset is highly relevant, but also substantially different from ours to be considered a complementary, rather than competing work.
>
> Thank you again for your time reviewing our paper. We appreciate your positive feedback and we hope that the newly provided statistics will improve the overall impression that our paper makes.

---

### Official Review · Reviewer_3cPB · 2023-08-11

**Soundness:** 3

**Excitement:**

3: Ambivalent: It has merits (e.g., it reports state-of-the-art results, the idea is nice), but there are key weaknesses (e.g., it describes incremental work), and it can significantly benefit from another round of revision. However, I won't object to accepting it if my co-reviewers champion it.

**Missing References:**

One obviously lacking comparison that is lacking is QuALITY (Wang et al., 2022). While the benchmark shares some differences with NarrativeXL, both aim to ostensibly address the same issue, which it the lack of long-document QA benchmarks

```
@inproceedings{pang-etal-2022-quality,
    title = "{Q}u{ALITY}: Question Answering with Long Input Texts, Yes!",
    author = "Pang, Richard Yuanzhe  and
      Parrish, Alicia  and
      Joshi, Nitish  and
      Nangia, Nikita  and
      Phang, Jason  and
      Chen, Angelica  and
      Padmakumar, Vishakh  and
      Ma, Johnny  and
      Thompson, Jana  and
      He, He  and
      Bowman, Samuel",
    booktitle = "Proceedings of the 2022 Conference of the North American Chapter of the Association for Computational Linguistics: Human Language Technologies",
    month = jul,
    year = "2022",
    address = "Seattle, United States",
    publisher = "Association for Computational Linguistics",
}
```

**Paper Topic And Main Contributions:**

The authors create a long-narrative QA dataset, NarrativeXL, aimed to be large enough to train LLMs. It also aims to provide better ways to evaluate how models perform in terms of long-term information retention. The authors provide results on GPT-3/4 to show the challenging nature of the benchmark.

**Questions For The Authors:**

- There is very little analysis of the dataset and the questions used within it. Is there a reason this is the case?
- The work argues for a number of limitations of NarrativeQA, so it would be very useful to have a comprehensive analysis of how NarrativeQA might fail to address things that NarrativeXL might from an empirical perspective. These experiments would appear to be critical for this paper to demonstrate it's worth, so why are these not included?
- Is there a comprehensive analysis comparing the quality, type and difficulty of questions here compared to those in other long document QA datasets? While the authors note that NarrativeXL is meant as a larger-scale alternative to NarrativeQA, if the questions are comparable in nature then there should be a proper quantitative analysis of where NarrativeXL shines in comparison.

**Reasons To Accept:**

- The problem is interesting and somewhat under-explored as of now.
- The benchmark could be useful for testing long-context LLMs.

**Reasons To Reject:**

While I appreciate the effort the authors put into the paper, I feel that there are a number of things that are lacking in this paper

- The work focuses very heavily on the NarrativeQA dataset, which is a comparatively old dataset for which many other alternatives have been presented and are not really discussed anywhere in this paper.
- The experiments somewhat lacking, with only one fine-tuned model (GPT-3). In particular, I would expect some long-context Transformers to be of interest here yet nothing is provided.
- The presentation of the paper appears very scattered and difficult to read. A great deal of the writing also appears to come from a somewhat speculative angle and do not seem necessary.

**Reproducibility:**

3: Could reproduce the results with some difficulty. The settings of parameters are underspecified or subjectively determined; the training/evaluation data are not widely available.

**Reviewer Confidence:**

3: Pretty sure, but there's a chance I missed something. Although I have a good feel for this area in general, I did not carefully check the paper's details, e.g., the math, experimental design, or novelty.

**Typos Grammar Style And Presentation Improvements:**

- Section 2 feels like it could be condensed as simply a single paragraphs as part of an analysis or background section. The paragraphs go into very subjective analysis which don't particularly add to the paper in any meaningful manner.
- Section 3 could be much better organized. Furthermore it focuses solely on NarrativeQA in a way that feels somewhat excessive and redundant with other parts of the paper.
- Section 5 would be much easier to understand with a figure or table which gives examples of the different question types.

- Tables 1 and 2 are somewhat dense in text and therefore difficult to parse.
- Table 3 is very difficult to read, especially the confidence intervals which make it very confusing to the reader what values are most important to compare.

---

> ### Author Rebuttal · Authors · 2023-08-29
>
> Thank you for your thoughtful review!
>
> First, we would like to address the general concern about us not training or fine-tuning more LLMs on our dataset. We agree that improving on state of the art LLMs would be ideal, but training a general purpose state of the art LLM on such long contexts would require tremendous resources, which we do not possess. We now acknowledge it in the limitations and clarify throughout our paper that we aim to provide a high-quality data source motivated by current LLM challenges, clearly stating that we hope rather than decisively prove that it would aid in SOTA LLM training.
>
> All of our experiments should be seen in this light – aiming to make sure that our data is of high quality (challenging, unique, and requiring long-term memory). We do not attempt to directly show that our data will improve SOTA LLMs, as we do not have the resources for it.
>
> In our revisions, we focused on this goal – making sure that it is clear how our dataset differs from existing resources and ensuring its quality.
>
> Below, we offer a point-by-point response to your comments and questions.
>
> __Q: “The work focuses very heavily on the NarrativeQA dataset, which is a comparatively old dataset for which many other alternatives have been presented and are not really discussed anywhere in this paper.”__
>
> A: We covered existing alternatives to the best of our ability (sections 3 and 7.2). In section 3, we purposely focused on the Narrative QA because, to the best of our knowledge, Narrative QA remains the most demanding dataset in terms of required memory length. The average document length in NarrativeQA is 50,000, while the closest dataset of comparable (though smaller) scale we could find was QMSum (Zhong et al., 2021), with 10,839 words on average.
>
> We apologize for missing the QuALITY (Wang et al., 2022) reference. We have now included it in the paper. Notably, however, despite some great properties, the QuALITY dataset offers a very different approach compared to our paper. Its data is carefully curated and hand-crafted, which is a plus, but the document length of 2 to 8k tokens and its size of 6,737 documents in total makes it less suitable for training or testing extreme-long-context memory LLMs (20k tokens and above).
>
> If there are references in the category of ultra long-context datasets that we have missed, we would be eager to include them, but we reported all we could find and, to the best of our knowledge, NarrativeQA remains the most relevant option in this category.
>
> __Q: “There is very little analysis of the dataset and the questions used within it. Is there a reason this is the case?”__
>
> A: Thank you for pointing this out. It was our oversight that we did not include a table listing thorough descriptive statistics of our dataset. We fixed it in the revised version. Please see the table in the general comment. We also include additional statistics in the appendix of our revised paper.
>
> When it comes to other analyses, given our procedure (generate scene and hierarchical summaries using GPT 3.5 -> generate questions based on them), all of our questions follow the same structure, which we described in the paper. Hence there is less uncertainty in our question structure than in datasets where questions or answers are human-generated. In our analyses, therefore, we focused on validating the quality of our questions (section 6).
>
> __Q: “The experiments somewhat lacking, with only one fine-tuned model (GPT-3). In particular, I would expect some long-context Transformers to be of interest here yet nothing is provided.”__
>
> A: Thank you for this suggestion. To strengthen our section 6.2, we conducted two additional fine-tuning experiments using Longformer Encoder-Decoder (base and large) models, fine-tuning them on the complete training set and evaluating on the test set.
> The results were as follows:
>
> LED base – Rouge1 F1: 0.52, Rouge2 F1: 0.26, RougeL F1: 0.40, BertScore F1: 0.90
>
> LED large – Rouge1 F1: 0.53, Rouge2 F1: 0.26, RougeL F1: 0.40, BertScore F1: 0.90
>
> Qualitatively and quantitatively, these results were close to the fine-tuned GPT3.5: the model learned to change the input in meaningful ways (changing some details in the scene summaries to their opposites), but still failed to outperform the baseline. Ultimately, this supports our previous conclusion that our summary reconstruction task requires long-term memory and does not appear to have simple shortcut solutions.
>
> __Q: “The work argues for a number of limitations of NarrativeQA, so it would be very useful to have a comprehensive analysis of how NarrativeQA might fail to address things that NarrativeXL might from an empirical perspective. These experiments would appear to be critical for this paper to demonstrate it's worth, so why are these not included?”__
>
> A: If we understand correctly, an empirical validation would be using our dataset to train/fine-tune a SOTA LLM, and then show that the improvement on some transfer task is higher than when we use Narrative QA. We agree that it is a possible approach, but we do not agree that it is the only approach, especially since NarrativeQA has a substantially different question structure from NarrativeXL (short QA questions in NarrativeQA vs mixture of multiple choice and paragraph-long freeform questions in our data). In these circumstances, different transfer tasks would likely benefit differently from NarrativeQA and NarrativeXL fine-tuning, almost guaranteeing an inconclusive result.
>
> Instead, we took a different approach and allocated substantial effort to show that our dataset __a)__ poses a challenge to modern language models (i.e. is not trivial to solve) __b)__ contains high-quality data __c)__ covers an area not covered by existing alternatives in terms of context lengths, scope, and question structure.
>
> Regrettably, we do agree that we could have performed better at “__c__”, since in our original submission, we did not have a summary table showing how our dataset differs from others. We apologize for that. After the rebuttal, we added a more clear descriptive statistics comparison with other datasets, including, among other options, Narrative QA and the recent QuALITY work by Wang et al. (see table in the general comment) showing that our dataset covers a gap in terms of dataset size and context lengths required to solve it.
>
> Overall, especially with this addition, we believe that we make a reasonably strong case that our dataset will be useful for the community.
>
> To be clear, we agree that a paper that would both propose a dataset and use it to improve the present state of Language Modeling would be ideal. But training or even fine-tuning and comprehensively evaluating state of the art LLMs demands extreme amounts of resources that we do not have. We now mention this fact in the limitations section.
>
> __Q: “Is there a comprehensive analysis comparing the quality, type and difficulty of questions here compared to those in other long document QA datasets? While the authors note that NarrativeXL is meant as a larger-scale alternative to NarrativeQA, if the questions are comparable in nature then there should be a proper quantitative analysis of where NarrativeXL shines in comparison.”__
>
> A: We have included descriptive statistics, highlighting the differences in scope and document length between our dataset and Narrative QA (please see our general comment). At the same time, while NarrativeQA shares similar goals with our work (long-range language understanding), the format of its questions is substantially different. In particular, Narrative QA is a Question Answering dataset, with answers being words or small phrases. In our case, the questions are either multiple choice (where the right scene summary has to be selected) or freeform (where a summary needs to be reconstructed). Therefore, a direct “side-by-side” comparison of question difficulty or quality is not quite possible.
>
> __Q: “Typos, grammar, and style comments''__ & __"The presentation of the paper appears very scattered and difficult to read. A great deal of the writing also appears to come from a somewhat speculative angle and do not seem necessary.”__
>
> A: Thank you very much for your paper organization suggestions! We have adjusted the paper to accommodate most of them. For example, we moved the last two points from section 3 to the appendix, as they are less central to the argument, and also reorganized Table 1 to be more readable. We have also moved the table with confidence intervals to the appendix. In the main paper, we now have a condensed version (no confidence intervals) and a reference to the full version in the appendix.
>
> As for question examples - to clarify, all multiple-choice read-along questions follow the same format as in Table 2, the difference is in how the “wrong” answer options are obtained. We clarified it in the paper and in the table description. We also included an additional illustration in the appendix to illustrate how different decoy answers were generated.
>
> Lastly, we went through the paper, aiming to scale down the speculative claims and to indicate more clearly when our statements are speculative.
>
> Thank you again for your time reviewing our article, as well as for your criticisms and suggestions. We hope that our rebuttal helps to address some of your concerns.

---

### Official Review · Reviewer_x1G6 · 2023-08-13

**Typos Grammar Style And Presentation Improvements:** N/A
**Soundness:** 3

**Excitement:**

3: Ambivalent: It has merits (e.g., it reports state-of-the-art results, the idea is nice), but there are key weaknesses (e.g., it describes incremental work), and it can significantly benefit from another round of revision. However, I won't object to accepting it if my co-reviewers champion it.

**Missing References:**

N/A

**Paper Topic And Main Contributions:**

The paper suggests a novel reading comprehension dataset designed for training and assessing long-term memory LLMs. The dataset creation involved a manual selection of books from Project Gutenberg, followed by the generation of scene summaries using the GPT-3.5 model. Subsequently, a set of multiple-choice questions was crafted to gauge the model's ability to comprehend text in a sequential manner. In order to assess the model's capacity for generating open-ended answers, the paper introduces questions that require corrections to end-of-book summaries. Lastly, the paper conducts various validation experiments to detail the dataset's benchmarks and inherent biases. The outcomes of these experiments underscore the substantial room for improvement in enhancing the performance of long-term memory LLMs.

**Questions For The Authors:**

Please refer to the reasons to reject section above.

**Reasons To Accept:**

1. The paper delves into a crucial facet of long-term memory within LLMs. For this purpose, the paper presents a novel dataset designed for the training and assessment of long-term memory capacities. In contrast to the currently available datasets, the paper provides a dataset that encompasses lengthier documents and a significantly larger number of examples. Furthermore, the dataset is intended for public release to facilitate future extensions and research endeavors.

2. The paper effectively demonstrates the considerable potential for enhancing long-term memory-driven advancements, given that prevailing state-of-the-art models struggle to differentiate accurate summaries generated by GPT-3.5 from the distorted summaries provided.


**Reasons To Reject:**

1. Interestingly, the paper omits concrete dataset statistics for the newly introduced datasets. Beyond the count of books employed in dataset generation, details such as the number of scenes, read-along/end-of-book summary questions, and other pertinent metrics are absent.

2. In the process of data generation, the paper employs the GPT-3.5 model to create both accurate and distorted summaries. To validate the precision of these summaries, a limited-scale human study was conducted, involving the enlistment of 5 workers who were assigned 10 scenes each. Out of a total of 50 scenes, 48 were correctly categorized. Given the emphasis on the extensive dataset size, can this modest study adequately substantiate the quality of the generated dataset?

3. In addition to validating and ensuring the suitability of the newly introduced dataset, the paper performs experiments using advanced models with extended context lengths, such as Cluade and GPT-4. It could have provided valuable insights to also include experiments involving publicly accessible state-of-the-art models, detailing their performance. This would have contributed to a more comprehensive grasp of the present status of long-term memory within LLMs.


**Reproducibility:**

3: Could reproduce the results with some difficulty. The settings of parameters are underspecified or subjectively determined; the training/evaluation data are not widely available.

**Reviewer Confidence:**

3: Pretty sure, but there's a chance I missed something. Although I have a good feel for this area in general, I did not carefully check the paper's details, e.g., the math, experimental design, or novelty.

---

> ### Author Rebuttal · Authors · 2023-08-29
>
> Thank you for your review and for your comments. We give a general overview of our rebuttal in our official comment. Below, we address your specific concerns point-by-point.
>
> Q1: “Interestingly, the paper omits concrete dataset statistics for the newly introduced datasets. Beyond the count of books employed in dataset generation, details such as the number of scenes, read-along/end-of-book summary questions, and other pertinent metrics are absent.”
>
> A: We deeply apologize for this, it was a very unfortunate oversight. We have now included such a table (please see the general comment). Hopefully it helps to better place our dataset in the context of existing research. We also include more detailed statistics in the appendix of our revised paper.
>
> Q2: “In the process of data generation, the paper employs the GPT-3.5 model to create both accurate and distorted summaries. To validate the precision of these summaries, a limited-scale human study was conducted, involving the enlistment of 5 workers who were assigned 10 scenes each. Out of a total of 50 scenes, 48 were correctly categorized. Given the emphasis on the extensive dataset size, can this modest study adequately substantiate the quality of the generated dataset?”
>
> A: Thank you for this question. We increased the sample size five-fold, now employing 25 workers, who were assigned 10 scenes each. Please note also that these are highly experienced MTurk users (master worker qualification + at least 1000 HITs before, with at least a 99% approval rate), so this data is of high quality.
>
> Out of 250 total scenes, the workers correctly identified 238, which gives a 95% binomial confidence interval of [0.92, 0.97] (each participant classified more than one scene, so using the binomial confidence interval is not strictly correct, but gives a reasonable idea). Overall, we believe that this new data gives sufficient confidence that our true and distorted summaries are adequately generated.
>
> It is also worth noting that summarization is, in general, a standard task for which GPT3.5 is known to have good performance, so, in a way, our approach is not pushing GPT3.5 far into untested waters.
>
> Q3: “In addition to validating and ensuring the suitability of the newly introduced dataset, the paper performs experiments using advanced models with extended context lengths, such as Cluade and GPT-4. It could have provided valuable insights to also include experiments involving publicly accessible state-of-the-art models, detailing their performance. This would have contributed to a more comprehensive grasp of the present status of long-term memory within LLMs.”
>
> A: Thank you for this comment! First, we would like to clarify that although we did use modern models, the primary goal of those experiments was to validate the quality of our data. Specifically, we wanted to ensure that memory indeed was crucial for good performance on our dataset, and that our task was not trivial even for the most advanced models. We did not aim to offer a comprehensive survey of modern model performance on our data, although our experiment could be used as a rough estimate.
>
> During the rebuttal, we added an additional experiment with Longformer Encoder Decoder (trying both the base and large versions), fine-tuning it in a context-less manner on our full dataset to further test for shortcut solutions.
>
> Results were as follows:
> LED base – Rouge1 F1: 0.52, Rouge2 F1: 0.26, RougeL F1: 0.40, BertScore F1: 0.90
> LED large – Rouge1 F1: 0.53, Rouge2 F1: 0.26, RougeL F1: 0.40, BertScore F1: 0.90
>
> The result will be added to section 6.2 and Table 3. Both models performed near baseline, slightly outperforming context-less GPT3.5 and GPT4. This reinforced our conclusion that summary reconstruction questions are robust against shortcut solutions and require context knowledge for good performance.
>
> Before submission, we had preliminarily tested a few larger openly available LLMs with context windows large enough for our task, but their zero-shot performance seemed unsatisfactory, while fine-tuning them required resources that we did not have. We, unfortunately, did not have enough time to fine-tune LED with context during the rebuttal period, but we will add its performance for completion for the camera-ready version (nothing crucial hinges on this result, so we hope that it’s okay for us to make this promise).
>
> Thank you again for your review and your suggestions. I hope that our answers resolve some of your concerns.

---

### Meta-Review · Area_Chair_QBnp · 2023-09-23

**Recommendation:** 3

**Metareview:**

As the invlved discussion with reviewrs shows, this is interesting and thought provoking work, however, there is consensus that the evalution as presented does not fully alleviate all the concerns of the reviewers.  The authors have been generous with the reviewers and put in the time to dilligently address the concerns raised, including imporving and expanding the human evaluation and the set of models tested, listing of data sets and more meaningful evaluation of the significance of their results.

I'd like to thank both the authors and reviwers for their engadement to improve this work.

---

### Decision · Program_Chairs · 2023-10-07

**Decision:**

Accept-Findings

**Comment:**

As the invlved discussion with reviewrs shows, this is interesting and thought provoking work, however, there is consensus that the evalution as presented does not fully alleviate all the concerns of the reviewers.  The authors have been generous with the reviewers and put in the time to dilligently address the concerns raised, including imporving and expanding the human evaluation and the set of models tested, listing of data sets and more meaningful evaluation of the significance of their results.

I'd like to thank both the authors and reviwers for their engadement to improve this work.